

# Epidemiology of kerosene poisoning in Saudi Arabia: a retrospective analysis

Bassam M. Hakami[1], Randa Mohammed Nooh[2] and Ali Ahmed Asiri[3]

[1] Field Epidemiology, Ministry of Health, Riyadh, Saudi Arabia
[2] Preventive Medicine & Public Health, Ministry of Health, Riyadh, Saudi Arabia
[3] General Directorate of Environmental Health, Ministry of Health, Riyadh, Saudi Arabia

## ABSTRACT

**Background:** Limited national studies exist on the epidemiology of kerosene poisoning in Saudi Arabia. This study aimed to determine the frequency, demographic distribution, geographic patterns, and seasonal variations of kerosene poisoning incidents in Saudi Arabia from January 2019 to December 2021.
**Methods:** This retrospective cross-sectional study utilized data on all reported kerosene poisoning cases from the National Poisoning Surveillance System. Cross-tabulation with chi-square tests assessed the relationships between poisoning cases and key variables such as gender, age group, and region.
**Results:** A total of 460 kerosene poisoning cases were documented: 32.2% in 2019, 37.2% in 2020, and 30.6% in 2021. Saudi nationals comprised 97.6% of cases, and the male populace felt more influenced (60.9%) than females (39.1%), although the variation that was found was not proven to be statistically significant ($p = 0.912$). Out of all age groups, kids between the ages of 1 and 5 were the ones to be primarily affected, accounting for 87.6% of cases ($p = 0.029$). Most incidents occurred in residential settings (83.7%) and involved oral ingestion (91.7%, $p < 0.001$). Regionally, the AlQrayat Region reported the highest number of cases (53%), followed by the Northern Borders (18%) and AlJouf (15.7%), with incidents peaking during the colder months. Hospital admissions accounted for 41.3% of cases, while discharges against medical advice (DAMA) increased notably from 8.1% in 2019 to 28.4% in 2021.
**Conclusion:** Kerosene poisoning in Saudi Arabia predominantly affects young children and occurs in residential settings, with higher concentrations in northern regions during colder months. Public health interventions focusing on parental education, safe kerosene storage practices, and region-specific prevention strategies are essential to reduce the burden of kerosene poisoning and improve outcomes.

## INTRODUCTION

Globally, unintentional poisoning remains a significant public health challenge. According to the 2022 World Health Statistics, unintentional poisoning caused 84,000 deaths worldwide, with rates ranging between 0.6 and 1.8 per 100,000 individuals (*World Health Organization, 2022*). This has contributed to the loss of approximately 5 million disability-adjusted life-years (DALYs) (*Khan et al., 2023*). As per the World Health

Corresponding author
Bassam M. Hakami,
bassam7hakami@gmail.com

Organization (WHO), it has been discovered that 91% of casualties due to unintentional accidents and 94% of DALYs are in low- and middle-income countries (*Chandran, Hyder & Peek-Asa, 2010*).

Children, especially those under the age of five, are the most vulnerable (*Ahmed et al., 2015*). The American Association of Poison Control Centers (AAPCC) revealed that in 2015 there were 1.3 million children who were exposed to poisonous substances and among them 40% were under 3 years of age (*Mowry et al., 2016*). In developing countries, pediatric poisoning often stems from household chemicals, with kerosene being a leading cause of unintentional poisoning, accounting for up to 70% of cases in pediatric emergency settings (*Prasadi et al., 2018*; *Oreh et al., 2023*).

Kerosene, a combustible oily liquid also known as paraffin or fuel no.1, is widely used for lighting, heating, and cooking in low-resource settings (*Maiyoh, Njoroge & Tuei, 2015*). Kerosene can cause severe health complications such as aspiration pneumonitis and, in extreme cases, death (*Lam et al., 2012*; *Kumar, Kavitha & Angurana, 2019*). Studies from Asia and Africa consistently show that children under 6 years of age are disproportionately affected by kerosene poisoning, largely due to unsafe storage practices, such as the use of unmarked or attractive containers, and a lack of adequate supervision (*Anwar et al., 2014*). Socioeconomic disadvantages and deficient parental education further exacerbate the risk (*Ahmed et al., 2022*; *Dayasiri, Jayamanne & Jayasinghe, 2017*; *Edelu et al., 2016*; *IK et al., 2021*; *Parekh & Gupta, 2017*).

In Saudi Arabia, kerosene continues to be used in many households, particularly in rural and colder regions. However, limited research has been conducted to assess the extent and characteristics of kerosene poisoning at the national level. Existing studies are either regional or general investigations of chemical poisoning (*Alshahrani et al., 2023*). For example, reports from Makkah and Jeddah highlight that a significant proportion of chemical poisoning cases involve children under five, most occurring in domestic settings (*Alnasser, 2022*; *Alzahrani et al., 2017*). However, no study has comprehensively investigated kerosene poisoning across the Kingdom using national data.

## Aim

The purpose of this research is to recognize the main factors that are relevant to kerosene intoxication in Saudi Arabia by making use of the data that the National Poisoning Surveillance System has between January 2019 and December 2021.

This study will provide the first national-level epidemiological assessment of kerosene poisoning in Saudi Arabia. The findings will help inform public health strategies to reduce poisoning incidents and promote safer practices in households across the Kingdom.

## MATERIALS AND METHODS

### Study design

This is a retrospective cross-sectional study investigating kerosene poisoning cases in Saudi Arabia as documented in the National Poisoning Surveillance System in the period from January 2019 to December 2021. The study specifically examined data during this 3-years period to ensure consistency, since considerable modifications were made to the

surveillance system from 2022 onwards. Utilizing available secondary data is a timely and low-cost approach that will enable us to investigate the prevalence and risk factors for kerosene poisoning in the Kingdom.

## Study population

The population of the study includes all the reported kerosene poisoning cases in the national poisoning surveillance system from January 2019 to December 2021. Each case of poisoning that is present to the emergency departments of any health facility over the Kingdom is reported and documented manually using the "Reporting Form for Chemical Poisoning or Drug over Dosage Poisoning". The forms recorded by the public health department in each facility and afterwards entered electronically onto the national surveillance system. The environmental health department in each health cluster reviews the data producing monthly reports, which are subsequently forwarded to the general department of environmental health at the Ministry of Health in Riyadh.

Data for this study was obtained from the National Poisoning Surveillance System at the General department of environmental health at the Ministry of Health. The collected data compiled using the standardized "Reporting Form for Chemical Poisoning or Drug over Dosage Poisoning" (Supplemental file S1). which includes a description of the individual patient's demographic characteristics, the type of exposure (which includes the method and place of poisoning), the treatments given, and the patient's results.

## Ethical considerations

The study was approved by the Ethical Review committee of the Ministry of Health in Saudi Arabia (IRB log No. 24-23 M approval date: 6/03/2024).

## Data analysis

To analyze the information, IBM SPSS Statistics version 27 (IBM Corp., Armonk, NY, USA) was used. Descriptive statistics were used to summarize the dataset. Categorical variables were displayed using the frequencies and percentages, and the continuous variables had means and standard deviations (SD).

To evaluate associations between kerosene poisoning cases (dependent variable) and independent variables such as gender, age groups, and region, the chi-square test of independence was applied. A $p$-value $< 0.05$ was considered statistically significant.

All 460 cases recorded in the National Poisoning Surveillance System between January 2019 and December 2021 were included in the analysis. Since the study utilized the complete dataset, no sampling, exclusion criteria, or sample size calculations were necessary.

## RESULTS

As presented in Table 1, Saudi citizens were most reported cases of kerosene poisoning (97.6%, $p = 0.657$). Males are more affected than women, but not significantly ($p = 0.912$). Most of the cases in the 1–5-year-old age group where the result of data ages was the most vulnerable group, which was the most significant thing found ($p = 0.029$). The monthly distribution of kerosene poisoning cases from 2019 to 2021 reveals a clear seasonal trend

**Table 1 Demographic distribution of kerosene poisoning cases (gender, age group, and nationality).**

| Variable | Categories | 2019 N = 148 (32.2%) | 2020 N = 171 (37.2%) | 2021 N = 141 (30.6%) | Total N = 460 | P value |
|---|---|---|---|---|---|---|
| Nationality | Saudi | 144 (97.3%) | 166 (97.1%) | 139 (98.6%) | 449 (97.6%) | 0.657 |
| | Non-Saudi | 4 (2.7%) | 5 (2.9%) | 2 (1.4%) | 11 (2.4%) | |
| Gender | FEMALE | 60 (40.5%) | 66 (38.6%) | 54 (38.3%) | 180 (39.1%) | 0.912 |
| | MALE | 88 (59.5%) | 105 (61.4%) | 87 (61.7%) | 280 (60.9%) | |
| Age group | <1 | 6 (4.1%) | 5 (2.9%) | 13 (9.2%) | 24 (5.2%) | 0.029 |
| | 1–5 | 135 (91.2%) | 15 3(89.5%) | 115 (81.6%) | 403 (87.6%) | |
| | 6–12 | 4 (2.7%) | 7 (4.1%) | 11 (7.8%) | 22 (4.8%) | |
| | 13–19 | 0 (0%) | 2 (1.2%) | 1 (0.7%) | 3 (0.7%) | |
| | 20–39 | 0 (0%) | 3 (1.8%) | 0 (0%) | 3 (0.7%) | |
| | >39 | 3 (2%) | 1 (0.6%) | 1 (0.7%) | 5 (1.1%) | |

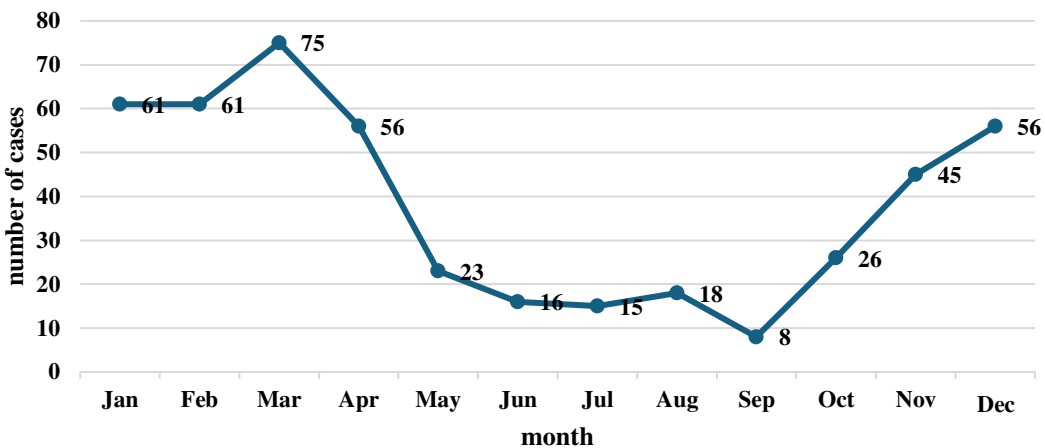

**Figure 1 Monthly distribution of kerosene poisoning cases (2019–2021).**

(Fig. 1), with a significant increase in cases during the colder months (January–April and December). The peak occurred in March with 75 cases, followed closely by January and February, each reporting 61 cases. After April, a sharp decline was observed, reaching the lowest point in September with only eight cases. A gradual rise began again in October (26 cases), continuing through November (45 cases) and December (56 cases).

The regional distribution of kerosene poisoning cases as shown in Fig. 2, and Table 2 highlights significant variation across Saudi Arabia ($p = 0.000$), with the AlQrayat Region consistently reporting the highest number of cases (53%, $n = 244$), peaking in 2020 (61.4%). The Northern Borders Region contributed 18% of cases ($n = 83$), while the AlJouf Region exhibited a dramatic increase, rising from 0% in 2019 to 38.3% in 2021 ($n = 72$). The Hail Region reported 7% of cases ($n = 32$), and the remaining areas, categorized as "Others," accounted for 5.2% ($n = 24$). The Riyadh Region consistently reported the fewest cases (1.1%, $n = 5$).

| Regions | Total |
|---|---|
| *AlQrayat Region* | 244 |
| *Northern Borders Region* | 83 |
| *AlJouf Region* | 72 |
| *Hail Region* | 32 |
| *Others* | 24 |
| *Riyadh Region* | 5 |

**Figure 2 Regional heatmap showing geographic distribution of kerosene poisoning cases.**

**Table 2 Regional distribution of kerosene poisoning cases.**

| Regions | 2019 N = 148 (32.2%) | 2020 N = 171 (37.2%) | 2021 N = 141 (30.6%) | Total N = 460 | P value |
|---|---|---|---|---|---|
| AlQrayat region | 85 (57.4%) | 105 (61.4%) | 54 (38.3%) | 244 (53%) | 0.000 |
| Northern borders region | 36 (24.3%) | 27 (15.8%) | 20 (14.2%) | 83 (18%) | |
| AlJouf region | 0 (0%) | 18 (10.5%) | 54 (38.3%) | 72 (15.7%) | |
| Hail region | 14 (9.5%) | 12 (7%) | 6 (4.3%) | 32 (7%) | |
| Others | 12 (8.1%) | 7 (4.1%) | 5 (3.5%) | 24 (5.2%) | |
| Riyadh region | 1 (0.7%) | 2 (1.2%) | 2 (1.4%) | 5 (1.1%) | |

Home is where most kerosene poisoning incidents occur (83.7%, $p = 0.000$), with the proportion increasing notably in 2020 (87.7%) and 2021 (84.4%), underscoring homes as the primary setting for exposure. Other locations, such as farms (0.4%) and schools (0.2%), were rare, while undefined locations accounted for 8.3% of cases (Table 3).

Regarding the route of exposure, oral ingestion was the predominant route (91.7%, $p = 0.000$), with the highest occurrence in 2020 (95.9%) and 2021 (95%). Inhalation and dermal exposure were minimal, contributing 0.9% and 0.2%, respectively, while undefined routes accounted for 7.2% (Table 3).

Management of kerosene poisoning cases showed significant variation over the study period ($p = 0.000$). Overall, 41.3% of cases required hospital admission, with the highest proportion in 2020 (48%). Cases with no admission accounted for 31.3% of the total, showing a slight decline in 2021 (28.4%). Notably, the proportion of cases discharged against medical advice (DAMA) increased from 8.1% in 2019 to 28.4% in 2021, totaling 17.6%. Rare outcomes included transfers (0.4%) and absconding (0.2%). The percentage of unidentified management outcomes decreased significantly from 27% in 2019 to 0% in 2021, (Table 4).

## DISCUSSION

This study explored the epidemiological characteristics of kerosene poisoning in Saudi Arabia from January 2019 to December 2021 using national surveillance data. Our findings

**Table 3  Place of incidence and route of exposure in kerosene poisoning cases (2019–2021).**

| Variable | Categories | 2019 N = 148 (32.2%) | 2020 N = 171 (37.2%) | 2021 N = 141 (30.6%) | Total N = 460 | P value |
|---|---|---|---|---|---|---|
| Place of incidence | Home | 116 (78.4%) | 150 (87.7%) | 119 (84.4%) | 385 (83.7%) | 0.000 |
|  | Farm | 2 (1.4%) | 0 (0%) | 0 (0%) | 2 (0.4%) |  |
|  | School | 1 (0.7%) | 0 (0%) | 0 (0%) | 1 (0.2%) |  |
|  | Other | 0 (0%) | 18 (10.5%) | 16 (11.3%) | 34 (7.4%) |  |
|  | Undefined | 29 (19.6%) | 3 (1.8%) | 6 (4.3%) | 38 (8.3%) |  |
| Route of exposure | Oral | 124 (83.8%) | 164 (95.9%) | 134 (95%) | 422 (91.7%) | 0.000 |
|  | Inhalation | 0 (0%) | 3 (1.8%) | 1 (0.7%) | 4 (0.9%) |  |
|  | Dermal | 0 (0%) | 1 (0.6%) | 0 (0%) | 1 (0.2%) |  |
|  | Undefined | 24 (16.2%) | 3 (1.8%) | 6 (4.3%) | 33 (7.2%) |  |

**Table 4  Management outcomes of kerosene poisoning cases (2019–2021).**

| Management | 2019 N = 148 (32.2%) | 2020 N = 171 (37.2%) | 2021 N = 141 (30.6%) | Total N = 460 | P value |
|---|---|---|---|---|---|
| Admission to hospital | 48 (32.4%) | 82 (48%) | 60 (42.6%) | 190 (41.3%) | 0.000 |
| No admission to hospital | 46 (31.1%) | 58 (33.9%) | 40 (28.4%) | 144 (31.3%) |  |
| DAMA | 12 (8.1%) | 29 (17%) | 40 (28.4%) | 81 (17.6%) |  |
| Transferred | 1 (0.7%) | 0 (0%) | 1 (0.7%) | 2 (0.4%) |  |
| Absconded | 1 (0.7%) | 0 (0%) | 0 (0%) | 1 (0.2%) |  |
| Unidentified | 40 (27%) | 2 (1.2%) | 0 (0%) | 42 (9.1%) |  |

align with global trends and highlight key demographic, seasonal, and geographic variations in kerosene poisoning cases, emphasizing the need for region-specific public health strategies.

Demographically, children aged 1–5 years were the most affected group, accounting for 87.6% of cases, a finding consistent with studies in Nigeria and South Asia that report toddlers' increased vulnerability due to hand-to-mouth behavior and curiosity-driven exposure (*Khan et al., 2023*; *Oreh et al., 2023*). Similar studies in Sri Lanka and Bangladesh confirm that young children are disproportionately impacted by kerosene poisoning (*Prasadi et al., 2018*; *Ahmed et al., 2022*). While males (60.9%) were more frequently affected than females, this gender disparity was not statistically significant ($p = 0.912$), suggesting that both genders remain at risk, aligning with findings from *Kumar, Kavitha & Angurana (2019)*.

Households were identified as the primary site of poisoning (83.7%, $p = 0.000$), corroborating global studies that emphasize the role of unsafe storage practices (*Chandran, Hyder & Peek-Asa, 2010*; *Alshahrani et al., 2023*). Ingestion of kerosene was the predominant route (91.7%), reflecting findings in Nigeria, India, and Sri Lanka, where

children often mistake kerosene for water or other consumables due to its storage in accessible or unlabeled containers (*Dayasiri, Jayamanne & Jayasinghe, 2017*; *Anwar et al., 2014*; *Maiyoh, Njoroge & Tuei, 2015*). These results highlight the urgent need for parental education on safe kerosene storage and child supervision.

Regional variations revealed a significant concentration of cases in the AlQrayat Region (53%), followed by the Northern Borders (18%) and AlJouf (15.7%), consistent with reports linking colder climates to increased kerosene use for heating (*Khan et al., 2023*). Local news and global studies confirm that demand for kerosene rises sharply during winter in northern regions, increasing the risk of poisoning incidents (*Lam et al., 2012*; *Alnasser, 2022*). This spatial pattern underscores the relationship between environmental factors and poisoning, necessitating region-specific interventions during winter months.

Seasonal trends demonstrated a clear peak in poisoning cases during January to April and December, aligning with the colder months when kerosene is extensively used for heating. These findings mirror results from studies in low- and middle-income countries (LMICs) such as Bangladesh, Sri Lanka, and India, where seasonal demand drives kerosene use (*Ahmed et al., 2022*; *Parekh & Gupta, 2017*). Such trends emphasize the need for preventive efforts, particularly during winter, including awareness campaigns and safer heating alternatives.

While this study provides valuable national-level insights, some limitations must be acknowledged. Data gaps regarding socioeconomic status, parental education, and storage conditions limited our ability to fully explore the underlying factors contributing to kerosene poisoning. Similar limitations have been reported in LMIC settings, where underreporting and incomplete data collection impede the accurate assessment of poisoning trends (*Oreh et al., 2023*; *Almutairi et al., 2023*). Future studies should address these gaps and explore the long-term health impacts of kerosene poisoning, particularly respiratory conditions like aspiration pneumonitis (*Balme et al., 2015*; *Tormoehlen, Tekulve & Nañagas, 2014*).

Despite these limitations, this study's strengths include its use of comprehensive, nationally representative surveillance data, providing a clear understanding of kerosene poisoning trends across Saudi Arabia. As the first national-level study on kerosene poisoning, it establishes essential baseline data to inform public health policies and interventions.

## CONCLUSIONS

This study provides the first national-level analysis of kerosene poisoning in Saudi Arabia, highlighting critical epidemiological patterns between 2019 and 2021. Children aged 1–5 years were identified as the most vulnerable group, with poisoning incidents occurring predominantly in households through oral ingestion. The AlQrayat Region and other northern areas reported the highest case concentrations, particularly during the colder months of January to April and December, reinforcing the seasonal association with kerosene use for heating.

These findings emphasize the need for targeted public health strategies, including parental education on safe kerosene storage, improved supervision of young children, and

region-specific interventions, particularly in high-risk areas. Awareness campaigns, safer heating alternatives, and policy-driven solutions are essential to mitigate the risks of kerosene poisoning.

While data gaps limited our ability to fully explore socioeconomic and environmental factors, this study establishes a robust baseline for future research and interventions. Addressing these limitations and exploring the long-term consequences of kerosene poisoning will further strengthen preventive measures and improve child safety in Saudi Arabia.

## ACKNOWLEDGEMENTS

We thank the Saudi Ministry of Health and Department of Environmental Health for providing access to the chemical poisoning surveillance data. Special thanks to all the staff in the Saudi field epidemiology training program for their continuous guidance and support throughout this study.

### Funding

The authors received no funding for this work.

### Competing Interests

The authors declare that they have no competing interests.

### Author Contributions

- Bassam M Hakami conceived and designed the experiments, performed the experiments, analyzed the data, prepared figures and/or tables, and approved the final draft.
- Randa Mohammed Nooh conceived and designed the experiments, authored or reviewed drafts of the article, supervision, and approved the final draft.
- Ali Ahmed Asiri conceived and designed the experiments, authored or reviewed drafts of the article, and approved the final draft.

### Ethics

The following information was supplied relating to ethical approvals (*i.e.*, approving body and any reference numbers):

The Institutional Review Board of Saudi Arabia's ministry of health (IRB log No.24-23 M approval date: 6/03/2024.

### Data Availability

The raw data is available in the Supplemental Files.

### Supplemental Information

Supplemental information for this article can be found online at http://dx.doi.org/10.7717/peerj.19094#supplemental-information.

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
