# Peer review of "Epidemiology of kerosene poisoning in Saudi Arabia: a retrospective analysis"

_PeerJ, doi:10.7717/peerj.19094_

## Round 0.1 · original submission · Major Revisions

The manuscript requires substantial revision. There are too many figures and tables, and most of them are redundant.
1. Separate the Methods into Data sources and Statistical Analysis
2. Figures 1-5 should be presented as a single table. Most of the tables are unnecessary and redundant.
3. Delete Figure 6 since you have Figure 7.
4. Where is the table of results for the inferential statistics?

·

Basic reporting

There is some major comments I recommend for improving the article and clarity.
The results not professionally represented. The raw data had many other information that not used like the outcome.
Background is too long but the references is good.
Self contained but the results need major revision.

Experimental design

Research questions fill the epidemiological gap only not the public health of kerosene poisoning.

Material & methods:
The surveillance needs to be detailed. How it was done manual or electronic. Inclusion criteria and exclusion criteria of the surveillance if there. Why does the surveillance take these 3 years? Does the surveillance take all the geographical area of Saudi Arabia? If yes the geographical regions covered by surveillance. The main points taken by surveillance.

Validity of the findings

The article is very important for Saudi Arabia but if we add the outcome it will affect the international burden of kerosene to compare in the next international studies.
Raw data had other available information other than the age, gender, and geographical variation that was not used.
Conclusions
The conclusions are well stated only for the epidemiological pattern not for the public health issue in the objectives in line 148.

Additional comments

Major revisions include
Abstract: they said that there is a correlation between occurrence of toxicity and age, gender and geography. Please instead of correlation using the statistically significant difference as chi-square and kruskal wallis do not measure correlation they measure difference.
Introduction is too long and not clear for the aim of the study. Why we start the article about the definition of injury and the difference between intentional and intentional. That will make the reader confused.
The introduction can start from line 65. From line 93 to line 106 they talked about the toxicological effect of kerosene in spite of no tables or figures representing the symptoms or signs of the cases in the study.
Line 134 one of the objectives of the study is talking about kerosine health effects, but in the article, there is no information about the mortality rate or complications rate like chemical pneumonia or the economic burden due to long hospital stays to direct preventable approach.
From line 137 to line 145 is a repeated point of the paragraph lines 130 to 136. No need to repeat it.
From line 146 to line 150 they can put a title of the aim of the study above this paragraph to be clear to the reader.
Material & methods:
The surveillance needs to be detailed. How it was done manual or electronic. Inclusion criteria and exclusion criteria of the surveillance if there. Why does the surveillance take these 3 years? Does the surveillance take all the geographical area of Saudi Arabia? If yes the geographical regions covered by surveillance. The main points taken by surveillance.
Results
The tables and figures are not professional, unfortunately. I suggest making a large table to represent the data difference in the 3 years and column for 2019, a column for 2020, a column for 2021 and a column for total cases and p-value of age, gender, the intention of poisoning, the incidence of chemical pneumonia, geographical distribution, hospital stay and outcome of the patients and a final column for the p-value difference in the 3 years.
In lines 242 to 246, the result of the test is not clear what are the median and quartiles of the compared data to get the Kruskal Wallis test? Kruskal Wallis test is used for numerical data represented in median and quartiles not for frequency variations as you said in line 244.
Keep graph 7 and add the frequency in a table in each month in the 3 different years with the p-value by chi-square test.
No need for Table 1 as you write it in the text. You can do the normality test for the age and according to its normality put it in the table I recommend you do this for the difference in the 3 years with the p-value.
Table two shows the age group differences in each year in addition to the total. P value in the column please write them professionally in the table.
Table 3 and Table 4 same comment as Table 2. I need the frequency in each year in addition to the total and p-value with the test you used.
Discussion
From lines 266 to 268 the paragraph is not clear and needs to be clarified in the tables in the result part.
Please remove the aim of the study in the discussion no need to repeat it. The lines 258 to 260.
Conclusions
The conclusions are well stated only for the epidemiological pattern not for the public health issue in the objectives in line 148.
Lines 336 to 345 put them under title limitation and recommendations.

·

Basic reporting

Dear Editor,
Thank you for the opportunity to review this article for your esteemed journal. I have carefully reviewed the submission and have found the following:
1. The English used throughout the article is clear, unambiguous, and professional.
2. The introduction provides sufficient background knowledge and includes appropriate references.
3. The article is structured according to the journal's guidelines.
4. The raw data has been shared.
5. The figures and tables are relevant to the content and are clearly visible and readable.
6. The results align with the stated aims and objectives.
Howeever following modifications needed
1. restrict the word count of the abstrct within 250 words according to the journals criteria.
2 provide the refrence for the sentence in the introduction part between lines 44-46.
3. reference number 6 is not matching with tthe data provided between the lines 56-61

Best regards,

Experimental design

According to the methodloly of the article, the experimental design is appropriate.

Validity of the findings

no comment

Additional comments

no comment

·

Basic reporting

The article meets the standard and there are only few corrections which I have pointed out on the PDF and require only some deletions and replacement. Actually I made the review on the PDF and I attached it.

Experimental design

No comment

Validity of the findings

The findings are valid and impact full to the community and worldwide. No comments

---

## Round 0.2 · Major Revisions

1. Present Table 5 as a time-series line plot and delete the current Figure 1. Present the whole data as a monthly time-series plot from 2019 to 2021
2. Combine Tables 2-4.
3. The data analysis needs to be rewritten to include all statistical techniques used in the study.

·

Basic reporting

First I want to thank authors as they made major revision but still there some issues I see should be done
Some parts are ambiguous like bivariate analysis in abstract what do you mean? and table 6 you not inform which is the dependent factor you depend to get that these independent factors can influence incidence of kerosene. please inform the test used to get this P value. If you used chi-square it is called crosstabulation and you should inform the groups you used to get the influence of these factors on it.
DAMA in table 4. Please any abbreviations in tables should be clear. You should make legend after any table for the test you used and any abbreviations in the table.
Table 1 is not professional where is the p value to compare the three year as regard age, sex and regions. Pleasr provide only five regions to compare not all these regions which is about 15 regions in table 1 to compare between the three years.
Results are not professional please seek for professional statistician to revise your results and to make inferential analysis and sample size calculation.

Experimental design

No comment

Validity of the findings

Statistics need to be revised clearly to be sound. They should provide good inferential analysis with power of the sample size.
Conclusions need to be revised after revision of the results.

·

Basic reporting

no comment

Experimental design

no comment

Validity of the findings

no comment

Additional comments

I appreciate the efforts made by the author(s).

---

## Round 0.3 · Major Revisions

I agree with the reviewers that more inferential statistics are essential to give more credence to your work. Since the study is based on national data, what is the spatial distribution? Incidence rates? Is there any seasonal variation (tested)?

·

Basic reporting

Introduction is good
Aim of the work is clear
But unfortunately not professional article structure, figures and tables
The section results is only descriptive which not attract the international reader. The previous two revision I suggest you to ask a professional medical statistician for his help but I see you don't.

Experimental design

research question is good but need to attract the international reader by adding inferential statistics
The paper only retrospective the data collected from previous study so if we don't add inferential statistics the paper will be weak.
You take the entire studied population but you can calculate the power of the sample size
No investigations were done

Validity of the findings

The paper not statistically sound, no novelty just descriptive statistics which don't give me a benefit as a reader.
I can't rely on conclusion as these descriptive not inferential statistics

Additional comments

Table 1 you compare between the 3 years as in gender the p value is the difference between the 3 years or between the gender as you didn't write any year have the difference so, I asked before about compare between each two years and totally.
where is the outcome?
if there is no mortality you can divide the outcome between long stay and short stay in hospital.

·

Basic reporting

.

Experimental design

.

Validity of the findings

.

Additional comments

I have re-reviewed the paper and advice that you can go ahead to accept and publish the article in its present form.

---

## Round 0.4 · accepted · Accept

All reviewer's comments have been adequately addressed.

·

Basic reporting

clear and the results section had improved significantly than before. Thanks very much

Experimental design

Good

Validity of the findings

Improved as the paper now concluded that 2020 had more kerosene cases due to that children and parents were restricted in home with parents business were done from home make increasing in pediatric poisoning cases in 2020 in general. That's sound internationally. many thanks

Additional comments

Thanks for improving table 1